# Serum Biomarkers in Carotid Artery Disease

**DOI:** 10.3390/diagnostics11112143

**Published:** 2021-11-18

**Authors:** Vassiliki I. Kigka, Vassiliki Potsika, Michalis Mantzaris, Vassilis Tsakanikas, Igor Koncar, Dimitrios I. Fotiadis

**Affiliations:** 1Unit of Medical Technology and Intelligent Information Systems, Department of Materials Science and Engineering, University of Ioannina, 45110 Ioannina, Greece; kigkavaso@gmail.com (V.I.K.); vpotsika@gmail.com (V.P.); mdmantzaris@gmail.com (M.M.); vasilistsakanikas@gmail.com (V.T.); 2Department of Vascular and Endovascular Surgery, Faculty of Medicine, University of Belgrade, 11000 Belgrade, Serbia; dr.koncar@gmail.com; 3Department of Vascular and Endovascular Surgery, Clinic Center of Serbia, 11000 Belgrade, Serbia; 4Institute of Molecular Biology and Biotechnology, Department of Biomedical Research Institute—FORTH, University Campus of Ioannina, 45110 Ioannina, Greece

**Keywords:** carotid artery disease, carotid stenosis, atherosclerosis, serum biomarkers, risk stratification, ultrasound, plaque vulnerability

## Abstract

Carotid artery disease is considered a major cause of strokes and there is a need for early disease detection and management. Although imaging techniques have been developed for the diagnosis of carotid artery disease and different imaging-based markers have been proposed for the characterization of atherosclerotic plaques, there is still need for a definition of high-risk plaques in asymptomatic patients who may benefit from surgical intervention. Measurement of circulating biomarkers is a promising method to assist in patient-specific disease management, but the lack of robust clinical evidence limits their use as a standard of care. The purpose of this review paper is to present circulating biomarkers related to carotid artery diagnosis and prognosis, which are mainly provided by statistical-based clinical studies. The result of our investigation showed that typical well-established inflammatory biomarkers and biomarkers related to patient lipid profiles are associated with carotid artery disease. In addition to this, more specialized types of biomarkers, such as endothelial and cell adhesion, matrix degrading, and metabolic biomarkers seem to be associated with different carotid artery disease outputs, assisting vascular specialists in selecting patients at high risk for stroke and in need of intervention.

## 1. Introduction

Carotid artery disease is a pathologic condition in which atherosclerotic plaques accumulate inside the carotid arteries, which are responsible for supplying blood to the brain. Carotid atherosclerosis is considered as a major cause of strokes, the third-leading cause of morbidity and mortality in the Western world, and among the leading causes of long-term disability in both men and women. Despite recent advances in medical and surgical management of carotid artery disease, the cerebrovascular burden of atherosclerosis remains high [1,2].

The degree of stenosis of carotid arteries, as determined by carotid artery imaging techniques, is considered the major risk factor for strokes and contributes to the selection of the management of patients with carotid atherosclerosis. More specifically, in patients with a symptomatic internal carotid artery (ICA) stenosis higher than 70%, carotid endarterectomy (CEA) is strongly recommended, whereas in individuals with cerebral ischemia and a carotid stenosis higher than 50%, as well as in asymptomatic individuals with a carotid stenosis over 70%, CEA or carotid artery stenting (CAS) should be discussed [3].

However, in some cases, the degree of stenosis alone is not enough for the selection of conservative or surgical treatment. Thus, except for the use of carotid imaging techniques, there is a need for complementary serum biomarkers that may contribute to the identification of high-risk carotid artery disease patients, or patients who are candidates for surgical intervention.

During the process of atherosclerotic plaque progression, specific molecules may diffuse toward the serum and provide information as circulating biomarkers of plaque presence, status, and risk of complications. The biomarkers which should be considered as surrogate for clinically significant cardiovascular endpoints should satisfy several criteria. According to the American Heart Association (AHA), the phases of evaluation of a surrogate biomarker for a cardiovascular endpoint include the proof of concept, prospective validation, incremental value, clinical utility, clinical outcomes, and cost effectiveness [4].

Different studies have presented serum biomarkers related to carotid artery disease [5,6]. This review aims to present all the available serum biomarkers with clinical significance for both the diagnosis of carotid artery disease and for its prognosis. More specifically, this review focuses on the biomarkers related to the presence of carotid artery stenosis, the vulnerability of plaques, and those associated with high risk of a stroke event and with high cardiovascular mortality.

## 2. Biomarkers Description

### 2.1. Inflammatory Biomarkers

Atherosclerosis is considered a chronic inflammatory disease, and inflammation plays a significant role in mediating all stages of atherosclerotic disease. Immune cell types (monocytes, macrophages, T-cells, and neutrophils) and specialized lipid mediators significantly contribute to vascular inflammation and are activated by risk factors present in the vascular wall, such as shear stress, oxidized lipoproteins, and oxidative stress [7].

A C-reactive protein (CRP) is considered one of the most significant biomarkers of inflammation, and the measurement of both CRP and high-sensitivity CRP (hs-CRP) is widely used in clinical practice for vascular disease stratification. Several studies have indicated that hs-CRP is associated with the presence of unstable carotid artery stenosis [8], the presence of ICA stenosis [9], and the detection of vascular risk patients [10]. Elevated levels of hs-CRP are associated with lower echogenicity of carotid plaques, suggesting a relation between the hs-CRP and the potential vulnerability of the plaques [11], whereas vulnerable atherosclerotic plaques indicated upregulation of hs-CRP [12]. Nevertheless, increased levels of hs-CRP are independently associated with a high risk of ischemic stroke [13], and among the risk factors for acute anterior circulation stroke [14].

Pentraxin 3 (PTX3), another acute phase protein, constitutes a potential inflammatory biomarker and is shown to be independently associated with the severity of carotid atherosclerosis [15,16]. Additionally, Shindo et al. investigated the prognostic significance of PTX3 of the vulnerability of carotid plaques through immunohistochemical analysis [12].

Interleukin-6 (IL-6), a pleiotropic proinflammatory cytokine, has been shown to be localized into inflammatory cells in vulnerable plaques, and its elevated levels are associated with a high risk of atherosclerotic plaques [11,12]. Additionally, IL-6 was shown to be associated with the presence of ICA stenosis [9], and through a genetic association study, it was indicated that IL-6 is associated with ICA stenosis [17].

Tumor necrosis factor-alpha (TNF-α), an inflammatory cytokine involved in early inflammatory events, has been correlated with the prevalence and severity of carotid artery stenosis [16] and with a high risk for carotid plaques. Vulnerable atherosclerotic plaque showed upregulation of the TNF-α, and increased levels of TNF-α are directly linked with high-size carotid plaques [12,17]. Immunohistochemistry analysis indicated that TNF-α, combined with hypoxia and oxidized LDL, markedly increased MMP-7 expression, which is directly associated with symptomatic carotid artery disease [18], whereas immunohistochemistry analysis of plaques after CEA indicated that TNF-α was significantly increased in symptomatic patients [19].

### 2.2. Endothelial and Cell Adhesion Biomarkers

Cell adhesion molecules (CAMs) are responsible for the regulation of the inflammatory response and endothelial function. Selectins, a family of cell-surface glycoproteins, are involved in the rolling and anchoring of leukocytes on the vascular wall, whereas intercellular adhesion molecules (ICAMs) and vascular cell adhesion molecules (VCAMs) induce firm adhesion of inflammatory cells at the vascular surface. Expression of VCAM-1, ICAM-1, and L-selectin has been consistently observed in atherosclerotic plaques and their soluble forms have been identified in the circulation [20].

More specifically, vascular cell adhesion protein 1, also known as a vascular cell adhesion molecule 1 (VCAM-1), showed a significant association with the ICA stenosis [9] and cardiovascular mortality [21]. As for high-risk plaques, vulnerable atherosclerotic plaques indicated upregulation of VCAM-1 [12]. ICAM-1, an endothelial- and leukocyte-associated transmembrane protein, also showed a significant association with cardiovascular mortality [21] and the presence of ICA stenosis [22].

Selectins, a family of cell adhesion molecules (or CAMs), have also been indicated to be associated with carotid artery disease, with E-selectin gene variants independently and significantly associated with ICA stenosis [22], whereas high-risk atherosclerotic plaques showed its upregulation [12]. On the other hand, lower values of L-selectin were associated with atherosclerotic plaque size [17].

Additionally, neutrophil gelatinase–associated lipocalin (NGAL) is found in granules of activated human neutrophils and has been proposed as a valuable biomarker for the detection of unstable carotid plaques in asymptomatic patients [23].

### 2.3. Matrix-Degrading or Proteolysis Biomarkers

Matrix metalloproteinases (MMPs) contribute to the degradation of both matrix and nonmatrix proteins, involved in the process of plaque destabilization and cap erosion. This function takes place in the extracellular environment. Different studies have indicated that MMPs play a significant role in the detection of vulnerable high-risk atherosclerotic plaques in patients with advanced carotid artery stenosis.

It was indicated that the combination of MMP-1, MMP-7, and TIMP-1 demonstrated the highest positive predictive value of 89.4% and a negative predictive value of 60.1% for patients correctly classified as individuals with unstable and stable carotid lesions by means of blood sample analysis [24]. Additionally, levels of MMP-9 were significantly elevated in individuals with unstable atherosclerotic plaques in comparison with those with stable ones [25], whereas immunohistochemistry analysis indicated that the mRNA levels of MMP-2, MMP-7, MMP-9, and MMP-14 were elevated in vulnerable plaques, among which expression of MMP-2 and MMP-14 were the highest [26]. In a similar study, Sigala et al. [27] concluded that MMP-9 is directly related to plaque instability.

In another direction, other studies attempted to correlate levels of MMPs with the presence of symptomatic carotid artery disease. Elevated levels of MMP-2 and MMP-9 were observed in patients with symptomatic carotid artery disease in comparison with those without symptoms [25]. On the other hand, Abbas et al. [18] concluded that carotid artery disease patients had significantly high plasma levels of MMP-7, compared with healthy individuals, with the highest levels of MMP-7 in patients with symptoms within the last 2 months [18]. ICA stenosis was shown through a genetic association study to be associated with MMP-3 and MMP-9 gene variants [22]. In addition to this, total mortality was also independently associated with elevated plasma levels of MMP-7 [18], whereas higher serum MMP-9 levels in the acute phase of ischemic stroke were associated with an increased risk of mortality and major disability [28].

### 2.4. Lipid Biomarkers

Lipid factors are, together with inflammatory factors, the main actors in the onset, evolution, and destabilization of atherosclerotic plaque. First, low density lipoprotein cholesterol (LDL-C) is independently related with the presence and extent of subclinical early systematic atherosclerosis [29]. Low levels of LDL-C are likely to prevent large artery atherosclerosis [30], could reduce the ischemic complications, and affects the plaque stability and antithrombus formation [31]. Other studies have shown that increased levels of LDL-C were independent risk factors for the occurrence of carotid stenosis [16] and showed positive associations with ischemic stroke [32].

The oxidation of LDL is considered a significant atherogenic modification of LDL within the vascular wall, where ox-LDL can trigger the expression of adhesion molecules on the cell surface, and thus stimulate the activation of endothelial cells. These adhesion molecules mediate the rolling and adhesion of blood leukocytes that adhere to the endothelium and migrate into the intima, causing the activation of macrophages, the release of proinflammatory cytokines, and the production of proteolytic enzymes, contributing to matrix degradation and plaque destabilization [33]. The role of circulating ox-LDL has gained considerable attention and low levels of ox-LDL are a promising therapeutic target against atherosclerosis [34]. In a study conducted by Sigala et al. [27], it was indicated that ox-LDL levels were associated with the presence of clinical symptoms of carotid artery disease. Additionally, there are efforts to clarify the correlation between the morphology of human atherosclerotic plaques and the ox-LDL levels in plasma and plaques. It was shown that elevated ox-LDL levels are related with a vulnerability to rupture [35]. Ox-LDL levels are also considered as significant biomarker for the prognosis of carotid artery disease. In a study by Markstad et al. [36], the authors indicated that ox-LDL leads to the release of sLOX-1 from endothelial cells and that circulating levels of sLOX-1 are associated with the risk of ischemic stroke, whereas Wang et al. [37] showed that elevated levels of ox-LDL were associated with the high risk of mortality and poor functional outcome within one year after stroke onset.

On the other hand, high density lipoprotein cholesterol (HDL-C) is characterized by its antioxidant, antithrombotic, anti-inflammatory, and antiapoptotic characteristics and may play a significant protective role in an acute stroke, protecting and limiting the ischaemia on the blood–brain barrier (BBB) and on the parenchymal cerebral compartment [38]. Lower levels of HDL-C were independently associated with an increased risk of having echolucent, rupture-prone atherosclerotic plaques [39,40,41]. Moreover, HDL-C contributes as a prognostic marker for the severity of stroke, since low baseline HDL-C (≤35 mg/dL) at admission was associated with higher stroke severity and poor clinical outcome during follow-up in patients with atherosclerotic ischemic stroke [42].

Triglyceride-rich lipoproteins (TRLs), a pool of lipoproteins that includes chylomicrons, very low density lipoproteins (VLDLs), intermediate density lipoproteins (IDLs), and other remnant lipid metabolism particles, appear to promote atherogenesis independently of LDL-C [43]. Elevated levels of TRLs seem to be associated with plaque echolucency, which is characterized by increased lipid content and macrophage density plaques. Echolucent carotid plaques were proven to be related with a higher risk for future ischemic stroke, particularly in previously symptomatic individuals, for restenosis after endarterectomy, as well as for myocardial infarction [41,44]. On the other hand, in a study presented by Kofoed et al. [44], it was shown that TRLs are elevated in patients with carotid artery stenosis higher than 50%, as compared with controls.

Although circulating lipoprotein-associated phospholipase A(2) (Lp-PLA2) has been considered as novel biomarker for cardiovascular diseases, its correlation between the atherosclerotic plaque expression of Lp-PLA2, inflammation, stability and the presence of clinical symptoms, especially for cerebrovascular disease, remains poorly defined. Nevertheless, in a study conducted by Mannheim et al. [45], it was shown that symptomatic carotid artery plaques are characterized by increased levels of Lp-PLA2, strongly supporting the role of Lp-PLA2 in the pathophysiology and clinical representation of cerebrovascular disease. In addition to this, Lp-PLA2 consists of a significant biomarker for the management of asymptomatic patients with carotid artery disease, since its elevated levels are directly associated with a high grade of carotid stenosis [46], and with the presence of unstable atherosclerotic plaques [14,46]. Regarding the prognostic significance of Lp-PLA2, it has been shown that its activity is an independent predictor for coronary heart disease and ischemic stroke in the general population [47] and Lp-PLA2 in its highest levels had an increased risk of recurrence after the first ischemic stroke [48].

Apolipoproteins (apos) are the protein components of plasma lipoproteins, which consist of a core of triglyceride and cholesterol esters, and a peripheral region of phospholipid, sphingolipid, and protein. The most relevant subtypes are considered the apo AI (the main protein of HDL), apo B-100 (the main protein of LDL), apo C-II (important in chylomicrons and VLDL, activates lipoprotein lipase), and apo E (present in chylomicrons, VLDL, and IDL, allowing for the binding of these lipoproteins to the hepatocytes).

ApoA-I (or apoA1) levels may be clinically useful for the diagnosis of stroke and for the differentiation between ischemic and hemorrhagic strokes [49], whereas reduced apoA-I levels are risk factors for a first ischemic stroke, and elevated apoA-I is considered a risk factor for a first hemorrhagic stroke [50]. A meta-analysis by Paternoster et al. [51] indicated a clear association of APOE with carotid intima media thickness (IMT), suggesting the possibility of a specific association with a large artery ischemic stroke.

Proprotein convertase subtilisin kexin type 9 (PCSK9) is a ptotease produced at the liver and is detectable in human plasma. It consists of a key regulator of the metabolism of LDL, and has recently been suggested to participate in the development of atherosclerosis [52]. Chan et al. [53] showed that serum PCSK9 levels are considered independent predictors of carotid IMT and may contribute to the increased risk of subclinical carotid atherosclerosis, independent of conventional risk factors. On the other hand, Xie et al. [54] concluded that plasma PCSK9 levels are associated with 10-year progression of atherosclerosis.

### 2.5. Metabolic Biomarkers

Proinflammatory chemerin, leptin, and resistin are considered as adipokines which influence vascular wall function. Association of circulating adipokines with cerebrovascular symptomatology and carotid plaque vulnerability were investigated. The results showed that low levels of chemerin and elevated levels of restinin were related to plaque instability, the risk of stroke, and the severity of carotid artery disease [55].

On the other hand, adiponectin, an anti-inflammatory and vasculoprotective adipokine, may act as a novel prognostic biomarker for atherosclerosis in stroke, since it was shown to be related with the risk of ischemic stroke [56,57]. In a study by Gustafsson et al., [58] the role of circulating adiponectin to vascular function and morphology was investigated. These authors concluded that elevated levels are associated with less arterial pathology, whereas Saarikoski et al. [59] supported the role of adiponectin in the pathophysiology of early atherosclerosis.

Fatty acid-binding protein 4 (FABP4) has been also shown to play an important role in macrophage cholesterol trafficking and has been considered as a key factor connecting vascular and lipid accumulation with the inflammation process. Increased levels of FABP4 are associated with the presence of carotid artery disease, plaque instability, and adverse outcome in patients with carotid atherosclerosis, since the highest mRNA levels of FABP4 have been observed in patients with the most recent symptoms [60,61].

Elevated levels of homocysteine (Hcy) have been associated with carotid plaque development and hyperhomocysteinemia has been described as an independent cardiovascular risk factor. More specifically, in a study presented by Alsulaimani et al. [62], it was shown that elevated Hcy was independently associated with plaque morphology and increased plaque area, subclinical markers of stroke risk, whereas high hyperhomocysteinemia prevalence in patients with extracranial cerebrovascular disease was confirmed by Alvarez et al. [63]. In addition to this, higher total Hcy levels were associated with asymptomatic carotid artery disease [64].

Osteoprotegerin (OPG) is a secretory glycoprotein which belongs to the tumor necrosis factor receptor family. its high concentration has been related with cardiovascular and vascular disease and contributes to atherosclerotic plaque stability. Studies have shown that higher levels of OPG have been observed in asymptomatic plaques related to symptomatic ones [65], and all plaques that exhibited calcification were significantly higher in asymptomatic patients. The effect of OPG in atherosclerotic plaques was confirmed by a study by Schiro et al. in which OPG was significantly elevated in symptomatic individuals related to the asymptomatic group [66].

## 3. Discussion

The purpose of this review study is to present the most significant biomarkers related to carotid artery disease and to identify their associations with clinically significant outputs of carotid artery disease. In Appendix A (in Appendix A), we present all the types of biomarkers related to carotid artery disease, including inflammatory, endothelial, and cell adhesion biomarkers, matrix degrading, and lipid related and metabolic biomarkers. We also indicate their associations with the presence of carotid artery stenosis, the diagnosis of vulnerability of carotid atherosclerotic plaques, and the presence of symptomatic carotid artery disease. Additionally, in Appendix A (in Appendix A) we present serum biomarkers with prognostic value for carotid artery disease, since they are related with a high risk of future stroke and high cardiovascular mortality.

CAS may potentially cause a debilitating stroke, and its accurate early detection is therefore important for the early diagnosis of carotid artery disease. In the literature, we investigate different studies aiming to identify the most significant serum biomarkers associated either with the presence of CAS greater than 50% [8,9,16,22,44], or with the early detection of subclinical atherosclerosis [10,59]. In these studies, the presence of carotid artery disease was mainly assessed by carotid ultrasound and the analysis of data was implemented using conventional statistical techniques, as is reported in Appendix A (in Appendix A).

Nevertheless, the focus of CAS diagnosis also shifted from pure stenosis quantification to plaque characterization, and more specifically to the detection of high-risk vulnerable plaques. This has led to further advancements in existing imaging tools and identification of high-risk plaque-related imaging characteristics, and to the identification of plaque histology characteristics related to high-risk plaques [67]. Appendix A (in Appendix A) indicates all the existing methodologies for the detection of vulnerable plaques. In most of these studies, the instability of atherosclerotic plaques was considered after the immunohistochemistry analysis of carotid plaques [12], whereas in other studies vulnerable plaques were considered after the implementation of a carotid ultrasound [17]. All types of serum biomarkers seem to be associated with the vulnerability of plaques and some of them have been also detected as useful biomarkers for the diagnosis of CAS, such as the hs-CRP, the PTX-3, the IL-6, and the TNF-a, concluding that the measurement of inflammatory biomarkers contributed both to the diagnosis and monitoring of the disease progression.

Additionally, inflammatory TNF-α, matrix-degrading biomarkers (MMP-2, MMP-7, MMP-9), ox-LDL, and metabolic biomarkers (restinin, osteopontin, osteoprotegerin) have been shown to be associated with symptomatic carotid artery disease, as presented in Appendix A (in Appendix A). The early detection of symptomatic carotid artery disease in patients may provide cost-effective disease detection and management strategies that can be used for symptomatic early–middle-aged adults.

Apart from the diagnostic-based circulating biomarkers in this review paper, biomarkers with a high-prognostic value for the progression of carotid artery disease are also reported. More specifically, biomarkers associated with the high risk of future stroke or cardiovascular event are described in Appendix A (in Appendix A). Circulating biomarkers which constitute the typical lipid patient profile (LDL-C, TC, triglyceride, Ox-LDL, Lp-PLA2), inflammatory biomarkers such as hs-CRP, TNF-α, IL-6, adipokines (adiponectin, leptin, FABP4) and homocysteine are related to the high risk of future stroke. Meta-analyses conducted by Gorgui et al. [62] and Gairolla et al. [63] concluded that levels of adiponectin and leptin are significantly associated with ischemic stroke, showing that adipokines may have a cause–effect relationship with carotid artery disease. As for cardiovascular mortality, typical hs-CRP biomarkers, endothelial and cell adhesion biomarkers (VCAM-1, ICAM-1), MMPs (MMP7, MMP-9), ox-LDL, and FABP4 seem to be associated with cardiovascular mortality, as shown in Appendix A. An indicative study by Zhong et al. [28] concluded that the high value of MMP-9 was associated with high cardiovascular mortality and major disability. In this study, data collection from 3186 participants were analyzed, with 767 of them having experienced major disability or death. In another study, Wang et al. [37] collected biochemical markers from 3688 patients and concluded that the patients in the highest ox-LDL quartile had a higher risk of 1 year stroke mortality.

The current review paper on circulating biomarkers which are related to carotid artery disease reported clinical significant biomarkers for the primary diagnosis of the disease and the patients’ risk stratification. In Appendix A (in Appendix A), we summarize all the presented studies based upon their participants and their clinical characteristics, providing specific biomarkers for the risk stratification of asymptomatic participants, symptomatic participants (either with a diagnosis of carotid artery disease or undergoing CEA), and symptomatic, asymptomatic, and control participants of the general population.

## 4. Conclusions

Heterogeneous data support the evidence that circulation biomarkers contribute to the more accurate assessment of the primary risk for a cerebrovascular event, the potential identification of high-risk asymptomatic patients, and to the management of asymptomatic carotid artery disease.

## Data Availability

Not applicable.

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
