# Peer review of "Serum Biomarkers in Carotid Artery Disease"

_diagnostics, 2021, doi:10.3390/diagnostics11112143_

Round 1

Reviewer 1 Report

please see the attached

Reviewer 2 Report

Article:

Serum Biomarkers in Carotid Artery Disease

 Abstract

Please remake (not clear): In this paper, our aim is to present all the available in the literature circulating biomarkers for the diagnosis and the prognosis of carotid artery disease. These biomarkers are derived by clinical based studies and statistical analyses.

Introduction

Please remake (not clear): Biomarkers to be considered as surrogate ones for clinical significant cardiovascular endpoints should satisfy several criteria, such as proof of concept, prospective validation, incremental value, clinical utility, clinical outcomes, cost-effectiveness, ease of use, methodological consensus, and reference values

Inflammatory biomarkers – please rewrite

The first two sentences.

? acute anterior circulation cerebral infarction

?Additionally, IL-6 showed to be  associated with the presence of ICA stenosis [9], and also through a genetic association study, it was indicated that IL-6 is associated with ICA stenosis [17].

These are just a couple of examples for not proper composition.

…………………..

The text is very rich in information, but the composition of sentences makes it very hard to be read. There is redundancy, the text is not properly organized, the tables are full with too more data.

Keeping in mind, that the topic is interesting for a large number of readers, I suggest rewriting the manuscript with careful English proofreading, to be more concise, more clear-cut, and the tables to be more simple and clear.

The discussion repeats the data presented previously.  Instead, the personal view of the authors regarding practical aspects (which biomarker to use, and when, in which patient), would have been welcome.

Round 2

Reviewer 2 Report

I think, the authors properly have solved the problems I mentioned in the review.